# The Impact of HPV Diagnosis and Abnormal Cervical Cytology Results on Sexual Dysfunction and Anxiety

**DOI:** 10.3390/ijerph20043630

**Published:** 2023-02-18

**Authors:** Seda Şahin Aker, Eser Ağar, Andrea Tinelli, Safak Hatirnaz, Fırat Ortaç

**Affiliations:** 1Department of Gynecology and Obstetrics, Gynecologic Oncology Division, Kayseri City Hospital, 38080 Kayseri, Turkey; 2Department of Vocational School of Health Services, Operating Room Services Division, Istanbul Gelisim University, 34570 Istanbul, Turkey; 3Department of Obstetrics and Gynecology and CERICSAL (CEntro di RIcercaClinicoSALentino), Verisdelli Ponti Hospital, 73020 Scorrano, Italy; 4Medical Park Hospital, In Vitro Fertilization Unit, 55200 Samsun, Turkey; 5Department of Gynecology and Obstetrics, Gynecologic Oncology Division, Ankara University, 06100 Ankara, Turkey

**Keywords:** cervical cancer screening, human papillomavirus, anxiety, psychological, sexual dysfunctions

## Abstract

Background: The objective of this study is to evaluate the effect of HPV diagnosis on the sexual function and anxiety levels of Turkish women. Methods: A total of 274 female patients who tested positive with HPV were included in the study and categorized into four groups: Group 1 (HPV 16/18 with normal cytology), Group 2 (HPV 16/18 with abnormal cytology), Group 3 (other high-strain HPV with normal cytology), and Group 4 (other high-strain HPV with abnormal cytology). All patients filled out the Beck Anxiety Inventory (BAI) and Female Sexual Function Index (FSFI) at the time when they tested positive for HPV and during the two-month and six-month follow-ups. Results: Significant increases were observed in BAI scores in all four groups, whereas significant decreases were observed in total FSFI scores in Groups 1 and 2 only(*p* < 0.05). BAI scores of Groups 1 and 2 were significantly higher than those of Groups 3 and 4 (*p* < 0.05). FSFI scores of Groups 1 and 2 measured during the sixth-month follow-up were significantly decreased (*p* = 0.004 and *p* < 0.001, respectively). Conclusions: Our findings suggest that patients with HPV 16 and 18 positivity and abnormal cytological findings are more likely to have high anxiety and sexual dysfunction.

## 1. Introduction

The human papillomavirus (HPV) infection is one of the most common sexually transmitted diseases observed in almost one quarter of women within the fertile period [1,2,3,4]. The diagnosis of atypical squamous cells or dysplasia is another common cytological abnormality detected in cervical cancer screening programs [5]. In addition to the positive high-risk strains of HPV 16 and 18 infections, colposcopic examinations are also required in the case of abnormal cytological findings. Being infected with a sexually transmitted virus and carrying the risk of contracting the diseases that may be caused by this virus such as, cervical intraepithelial neoplasia (CIN) and cervical cancer, are all factors that may cause emotional stress in infected women [1,4,6]. The adverse sexual and carcinogenic outcomes of HPV infection may lead to psychosocial disorders [4,7]. Subclinical anxiety or depressive symptoms have been associated with fear of cancer [1,8]. It has been reported that sexual and reproductive complications, including sexual difficulty or dysfunction, were observed in women who were otherwise sexually active after the diagnosis of HPV infection or an abnormal Papanicolaou smear test result [1,7]. It was speculated that all these factors might cause women to feel less sexually attractive and not to be satisfied with sexual intercourse [7,9,10].

It has been reported that psychological well-being may be more of a factor than physical well-being in patients infected with HPV infection [1,6,11]. Previous studies have focused on the negative impact of HPV infection on women’s psychological, social and sexual lives [1,4]. Female patients who receive positive HPV test results are reported to experience anxiety and worry [8]. Similar to this, Lin et al.’s study revealed that HPV patients encounter a number of unfavorable feelings, primarily dread and anxiety [12]. Another important point to be emphasized about HPV transmission is the negative effects on female sexuality. Ferenidau et al. reported that after being diagnosed with the HPV infection, women had less sexual interest and had less sexual intercourse [13]. Similarly, in the study of Lin et al., interviewing women infected with high-risk HPV genotypes revealed that these women reported that they had infrequent sexual intercourse after the HPV diagnosis [12]. Likewise, McCaffery et al. reported that women diagnosed with HPV hesitate to share their test results with their partners and have negative feelings about their sexual life. This may be due to fear of infecting the partner, fear of rejection, guilt, and anger at having been betrayed [8]. The results of the studies available in the literature on the effect of having tested positive for HPV on women’s sexual function and psychosexual condition are contradictory [1,4,7,8,9]. These discrepancies in results might be attributed to the differences in the methodologies of these studies, as well as to the differences in the cultural, social, and religious characteristics of the study samples. In the light of the foregoing, it was hypothesized that women who were infected with HPV 16 and HPV 18 strains had higher anxiety and lower sexual function levels. In this context, the aim of this study is to investigate the effects of different HPV types and cervical cytology results on the sexual functions and anxiety levels of Turkish women. Furthermore, we wanted to determine which subgroup of sexual functions is mostly affected by different HPV genotypes and cervical cytology results.

## 2. Materials and Methods

### 2.1. Research Design

This study was designed as a prospective observational study to evaluate the effect of an HPV diagnosis, HPV types, and cervical cytology abnormalities on the sexual function and anxiety levels of Turkish women.

### 2.2. Population and Sample

The study comprised all female patients who tested positive for HPV using the Papanicolaou [13] test plus liquid-based cervical cytology within a community-based national cervical cancer screening program in a tertiary university hospital between July 2020 and December 2021. The HPV genotyping and cytological evaluation of the Pap test were performed. For high-risk HPV (hr-HPV) genotyping, the Cobas 4800 system was used in accordance with the manufacturer’s instructions (Roche Molecular Systems, Alameda, CA, USA). Positive test results were classified as HPV 16, HPV 18 and “other hr-HPVs”, namely, HPV 31, 33, 35, 39, 45, 51, 52, 56, 58, 59, 66, and 68.

The majority of women were reached through community health centers, and HPV testing and cervical cytology were applied free of charge in all public hospitals. The national cervical cancer screening program includes only women who are ≥30 years old. Thus, patients under 30 years old were excluded from the study [4,14]. Additionally, only the sexually active patients, who were in their reproductive period, were included in the study. Patients who were in pregnancy, lactation, and postmenopausal periods, patients with a history of anogenital warts or HPV infection, patients with genital pathologies affecting sexual function such as an anogenital warts, or chronic systemic diseases such as diabetes mellitus, patients who were on antidepressants, patients who had been medically treated for psychiatric disorders, and patients with a current diagnosis of cervical cancer were excluded from the study. The medical records of the patients included in the study were reviewed in detail.

All patients included in the study were given an information sheet about the details of the study. The demographic and clinical characteristics of the patients, including age, gravida, contraception method, and smoking history, were recorded. The cytological findings were grouped as normal and abnormal cytology (low-grade intraepithelial squamous lesion (LG-SIL), high-grade intraepithelial squamous lesion (HG-SIL), and atypical squamous cells of undetermined significance (ASC-US)) [15,16]. All patients filled out the Beck Anxiety Inventory (BAI) and the Female Sexual Function Index (FSFI) at the start of the study [1,4]. The study sample was categorized into four groups based on the HPV genotypes and cytology findings: (a) HPV 16/18-positive with normal cytology (Group 1); (b) HPV 16/18-positive with abnormal cytology (Group 2); (c) Other-hr HPV-positive with normal cytology (Group 3); and (d) Other-hr HPV-positive with abnormal cytology (Group 4) [4]. All patients were informed about the results of the HPV test and the cytology results. The guidelines of the Turkish national cervical cancer screening program were followed [4,14]. Accordingly, all patients in Group 3 were referred to have a colposcopic examination as part of the institutional policy to check for the presence of abnormal cervical imaging or post-coital bleeding. The patients who were diagnosed with CIN 2 and CIN 3 were referred to have the loop electrosurgical excision (LEEP) procedure. The patients diagnosed with cervical cancer (*n* = 6) were excluded from the study. The patients’ psychological status was assessed three times: when they were notified about their HPV infection (baseline), when they were informed about the results of the colposcopic biopsy, when the decision to proceed to LEEP was made (the two-month follow-up), and during the six-month follow-up (Figure 1).

All patients underwent follow-up examinations and filled out the BAI and the FSFI during the two-month and six-month follow-ups. 

### 2.3. Beck Anxiety Inventory

The Beck Anxiety Inventory (BAI), which is a brief assessment of anxiety with an emphasis on somatic symptoms, consists of 21 questions. The Turkish validation studies of BAI have been completed [17,18]. BAI features a four-point Likert-type scale with choices ranging from zero (not at all) to 3 (severely). Accordingly, the total BAI score ranges between zero and 63. The higher the BAI total score, the higher the degree of anxiety. The BAI shows a high internal consistency (Cronbach’s alpha = 0.93). The item total correlations ranged from 0.45 to 0.72 [18]. 

### 2.4. Female Sexual Function Index

The Female Sexual Function Index (FSFI), a self-administered inventory, was used to quantify the sexual functions of the patients included in the study. The Turkish validation studies of FSFI have been completed [4,19]. FSFI contains a total of 19 items categorized into six subscales: desire, arousal, lubrication, orgasm, satisfaction, and pain. The items in the arousal, lubrication, orgasm, and pain subscales are assigned scores between 0 and 5, whereas the items in the satisfaction and desire subscales are assigned scores between 1 and 5. Accordingly, the total FSFI score ranges between 2 and 36. The higher the total FSFI score, the better the sexual function [4]. The internal consistency of the FSFI test is high, (Cronbach’s alpha = 0.83–0.96) and the test–retest reliability is ranged from 0.74 to 0.86 [19].

### 2.5. Statistical Analysis

The study’s primary outcomes were the BAI and total FSFI scores and the changes observed in these scores over time between the groups categorized based on the HPV status. The descriptive data were expressed as mean ± standard deviation values in the case of continuous variables that were determined to conform to the normal distribution and as median values along with minimum–maximum values in the case of continuous variables that were determined not to conform to the normal distribution. The categorical data were expressed as numbers and percentages. The Shapiro–Wilk, Kolmogorov–Smirnov, and Anderson–Darling tests were used to determine whether the numerical variables conformed to the normal distribution.

In the analysis of more than two independent groups, Pearson’s chi-squared test and Fisher’s exact test were used to compare the categorical variables between the groups. The Kruskal–Wallis test was used to compare the median BAI, total FSFI, and FSFI subscale scores. The Dwass–Steel–Critchlow–Fligner test was used to evaluate the differences between the groups. The study groups were re-categorized according to the HPV status of the patients as HPV 16/18 and other-hr HPV to clarify the effect of the HPV status on the scores. The Mann–Whitney U test was used to compare the median BAI, total FSFI, and FSFI subscale scores between two independent groups with numerical variables determined not to conform to the normal distribution. The relationship between the BAI scores and the changes observed in the total FSFI scores over time between the groups were evaluated by the non-parametric covariance analysis using the “sm. ANCOVA” package in the R software. The Spearman rho correlation coefficient was used to analyze the correlations between BAI and total FSFI scores. “Jamovi project (Jamovi, version 2.2.5.0, 2022, retrieved from https://www.jamovi.org, accessed on 17 February 2023), JASP software package (Jeffreys’s Amazing Statistics Program, version 0.16, Amsterdam, The Netherlands retrieved from https://jasp-stats.org, accessed on 17 February 2023), and R-project (version 4.1.2) were used in the statistical analysis. Probability (*p*) values of ≤0.5 were deemed to indicate statistical significance.

## 3. Results

There were 274 patients in the study. The number of patients in each group was as follows: 100 (36.5%) in Group 1, 57 (20.8%) in Group 2, 74 (27.0%) in Group 3, and 43 (15.7%) in Group 4. Twenty (20.0%) and 19 (33.3%) patients in Groups 1 and 2 were infected with multiple HPV genotypes, including other-hr HPV strains, respectively. The distribution of patients’ HPV genotypes, cytological and pathological findings, and the need for the LEEP procedure are shown in Table 1.

The distribution of patients’ demographic and clinical characteristics by study group is shown in Table 2. 

The groups were similar in age, gravida, contraception use, and methods (*p* > 0.05). There was a significant difference in the rate of patients who were active smokers between Group 1 and Group 2 (*p* = 0.029). The distribution of patients’ BAI and FSFI scores by study group is shown in Table 3. There was no significant difference in baseline BAI and FSFI scores between the groups (*p* > 0.05). In contrast, there were significant differences between the groups in terms of BAI and FSFI scores determined during the two-month and six-month follow-ups and the changes in BAI and FSFI scores over time (Figure 2 and Figure 3). 

In Figure 2, it can be seen that there were significant increases in the BAI scores over time compared to the baselines scores in all groups (*p* = 0.001 for all). The patients in Groups 1 and 2 had significantly higher BAI scores measured during the second-month follow-up than those in Groups 3 and 4 (*p* < 0.05). There were no significant differences between Group 1 and Group 2 and between Group 3 and 4 in BAI scores measured during the two-month follow-up (*p* > 0.05). Likewise, as can be seen in Figure 2, the patients in Group 2 had significantly higher BAI scores measured during the six-month follow-up than the patients in other groups (*p* < 0.001 for all cases). Additionally, the patients in Group 1 had significantly higher BAI scores measured during the six-month follow-up than those in Group 3 and Group 4 (*p* < 0.001 for both cases). There was no significant difference between Group 3 and Group 4 in BAI scores measured during the six-month follow-up (*p* = 0.983). The median increases in BAI scores between the two-month and six-month follow-ups were higher in Groups 2 and 1 than those in Groups 3 and 4 (Table 3, Figure 2).

As can be seen in Figure 3, there were significant progressive decreases in patients’ total FSFI scores over time compared to the baseline scores in Groups 1 and 2 (*p* < 0.001 and *p* < 0001). There were no significant differences between the groups in total FSFI scores determined during the second-month follow-up (*p* = 0.181). Figure 3 shows that the total FSFI scores determined in the six-month follow-up of the patients in Group 1 and 2 were significantly lower than those in Group 3; the patients in Group 1 and 2 had significantly lower total FSFI scores determined during the six-month follow-up than those in Group 3 (*p* = 0.004 and *p* < 0.001, respectively). The total FSFI score determined during the six-month follow-up was lower in Group 2 than in Group 1, albeit not statistically significant (*p* = 0521). The median percent decrease in the total FSFI score determined during the six-month follow-up compared to the total FSFI scores determined before was higher in Group 2 than in the other groups (Table 3, Figure 3). 

The covariance analysis of BAI scores revealed significant differences in the total FSFI scores determined during the six-month follow-up between Group 3 and Groups 1 and 2, indicating the differences in the anxiety levels (*p* < 0.001). 

There was no significant difference between the groups created based on HPV 16/18 positivity in baseline BAI and total FSFI scores and the total FSFI scores determined during the two-month follow-up (Table 4). 

The patients with HPV 16/18 positivity had significantly higher BAI scores determined during the two-month and six-month follow-ups than those with other-hr HPV results (*p* < 0.001 for both cases). The total FSFI scores determined during the six-month follow-up were significantly lower in patients with HPV 16/18 positivity (*p* < 0.001).

There were significant correlations between the groups in BAI and total FSFI scores (Table 5). 

There were significant correlations between the groups in BAI and total FSFI scores determined during the six-month follow-up in the negative direction. There were also significant correlations between the groups created based on HPV 16/18 positivity and abnormal cytology findings in BAI and total FSFI scores determined during the six-month follow-up in the negative direction (*p* < 0.05 for all cases).

## 4. Discussion

The study findings revealed that the patients who tested positive for HPV 16 and 18 had a higher degree of anxiety than the patients with other-hr HPV positivity. Additionally, it was observed that the anxiety levels of patients with HPV 16/18 positivity and abnormal cytological findings increased as the time elapsed to obtain the final results of consecutive diagnostic interventions was prolonged. The effect of the HPV grouping on sexual dysfunction was not as evident as its effect on anxiety. Patients with HPV 16/18 positivity had lower FSFI scores determined during the six-month follow-up compared to the patients with only other-hr HPV positivity and normal cytological findings. 

The results of the studies on the reciprocal relationships between HPV infection, anxiety, depressive symptoms, and sexual dysfunction are controversial. Patients with HPV infection may display different psychological reactions depending on the diagnosis and treatment of the infection [1]. Denial of HPV infection, fear of cancer, reluctance to engage in sexual activity and anxious and depressive symptoms are among the common reactions reported in these patients [1,5,9,12]. Several authors speculated that genital lesions might also develop in pre-cancerous HPV patients, leading to increased anxiety and sexual dysfunction [20]. In addition to the fear of cancer, sexual health, and the relationship with husband or partner are among the other concerns in these patients [1,20]. Therefore, it can be speculated that women infected with HPV will likely have sexual dysfunction to some extent. 

There are conflicting results about the effect of HPV infection on sexual function [1,4,7,8,9]. Compared to the patients with other sexually transmitted viral diseases, such as AIDS (acquired immune deficiency syndrome) caused by the HIV (human immunodeficiency virus) or HSV (herpes simplex virus) infections, patients with genital HPV were found to have better self-reported outcomes [21]. As a matter of fact, Reed et al. [22] did not find any significant difference in the psychosexual characteristics and functioning between women with and without HPV infection. Similarly, in a prospective, cross-sectional study, it was concluded that the sexual function was not related to the diagnosis of HPV infection or the time required elapsed until the diagnosis of HPV [9]. However, patients with abnormal cytological findings and anogenital lesions were not included in the said study. There are other studies where no correlation was found between HPV diagnosis and sexual dysfunction [4]. In contrast, Mercan et al. [1] found a significant correlation between sexual dysfunction and HPV diagnosis. In parallel, Nahidi et al. [7] showed that previously treated anogenital warts decreased marital satisfaction. Similarly, a significant decrease was found in the FSFI scores in all subgroups in this study. Namely, the FSFI score of Group 2 determined during the six-month follow-up was significantly different than that of other groups. Additionally, the median percent decrease in the FSFI score of Group 2 was also significantly more than the other groups. These results suggest that HPV 16/18 positivity and abnormal cytological findings exacerbate sexual dysfunction. 

In contrast, a study conducted in Norway reported that women’s anxiety and depression scores were not affected when the primary screening method was changed from cytology to hr-HPV testing [23]. On the other hand, several studies reported that abnormal cytological findings alone significantly increased anxiety levels, which, however, decreased over time [24]. In addition, it was reported that adverse social and psychological consequences were more common in patients who were notified about a positive HPV test result than in patients who were notified only about an abnormal smear [8]. Based on these findings, the higher sexual dysfunction levels observed in Group 2 during the six-month follow-up may be attributed to the synergistic effects of abnormal cytological findings. Therefore, it can be speculated that the increases in the uncertainty about the prognosis of both HPV infection and abnormal cytology and the prolongation of the time for the completion of the diagnostic procedures lead to higher levels of sexual dysfunction.

The effect of the diagnosis and the treatment of associated cervical intraepithelial neoplasia were studied in several studies [25]. In one of these studies, Mercan et al. [1] evaluated HPV-positive patients before the treatment process. In comparison, in this study, HPV-positive patients were evaluated in terms of the diagnostic procedures used and followed up for six months. The LEEP procedure was performed in 35.8% of the cases. The LEEP procedure was performed under local anesthesia as stated in Comba et al.’s study to eliminate the negative impact on patients’ pain experience and sexual functions [26]. Six patients who had cervical cancer were excluded from the study. In this way, the effect of the diagnosis and the treatment of cervical cancer diagnosis on anxiety and sexual dysfunction were attempted to be ruled out. It was reported in previous studies that the spontaneous interest of women who were referred for colposcopy experienced in sex was significantly less and that the frequency of intercourse in these women had significantly decreased after the colposcopy [27]. In contrast, no significant relationship was found between cervical epithelial dysplasia, LEEP, and sexual dysfunction in other studies [28,29]. Thus, the effect of colposcopy, LEEP, or abnormal cytology on sexual function remains controversial. 

In this study, not only the initial diagnosis of HPV infection but also the associated diagnostic procedures, including colposcopy and LEEP in selected cases, were investigated in terms of any correlation with anxiety and sexual dysfunction. In this context, it is important to consider the differences between the methodologies employed by the studies when comparing their results on sexual dysfunction following HPV infection. 

Alay et al. reported the outcomes of their cohort, including the patients with hr-HPV infection, within the first two-month follow-up period. The methodology of Alay et al.’s study [4] was conducted with a cohort including patients with hr-HPV infection during a two-month follow-up period following diagnosis, and the methodology of this study showed remarkable similarity, except for the longer follow-up period employed in this study. However, contrary to the results of this study, Alay et al. did not find any significant difference between the groups in total FSFI scores determined during the two-month follow-up. Thus, prospective studies with larger sample sizes and extended follow-up periods are needed. 

The relationship between sexually transmitted infections and adverse psychological outcomes has been addressed in several studies. In one of these studies, Leite et al. [30] found that HPV diagnosis had a significant psychosocial effect on the quality of life of HPV patients, which was manifested as depressive and anxious symptoms, sexual dissatisfaction, and emotional suppression. Nevertheless, they did not find any significant difference between the low-risk and high-risk HPV infection groups in terms of the said symptoms. Increased prevalence of depression or depressive symptoms and anxiety were reported in HPV-positive women [8,31,32,33,34]. In addition, HPV 16/18 infection was found to be more significantly associated with poorer quality of life than infections with other strains of HPV [10]. Furthermore, it was reported that anxiety was more common than depressive symptoms in HPV-positive patients [8,31]. The reason why no relationship was found between HPV infection and anxiety in some studies could be the relatively shorter follow-up periods employed in these studies [1,4]. On the other hand, Mercan et al. [1] reported that the impairment of sexual function was not related to depression or anxiety. Similar findings have been reported by others [4]. In contrast, the findings of this study collected during the two-month and six-month follow-ups revealed that anxiety levels significantly affected the sexual dysfunction levels of patients with hr-HPV infections. The psychosocial effects of HPV infection and abnormal cytological findings are not clear [35]. Kwan et al. [5] reported that HPV test results did not have any effect on the anxiety levels of patients with abnormal cytology. However, the psychosocial burden of HPV-positive patients was found to be high in the said study as in this study. Kwan et al. also reported that the initially heightened anxiety levels and cervical cancer worries decreased after colposcopy. The discrepancies in the results of the relevant studies might be attributed to the differences in the cultural characteristics of the study cohorts, the use of different time points for the measurements, the use of different tests to measure anxiety or other psychological stress-related factors [8]. Apart from its strengths, such as the longer follow-up period, which enabled the collection of more extensive data about the long-term effects of HPV positivity, there were also some limitations to this study. First, psychiatric interviews to evaluate anxiety were not conducted. Instead, BAI was used to determine the anxiety levels of patients. Secondly, anxiety and sexual function parameters were evaluated using only one measurement tool for each of the said parameters. If there were more cases included in our study, we could have obtained more statistically discriminating results. We could follow the patients for more than 6 months. However, due to the conditions in our country, we have difficulty in reaching all of these cases whenever we want. If we had extended it to 1 year, we would have had problems with the continuity of the patients. It is controversial whether the LEEP procedure has an effect on sexual functions. The fact that the LEEP procedure performed in the HPV 16–18 group was higher than the other group may be a negative impact factor for sexual dysfunctions (43% of Group 1 and Group 2; 24% of Group 3 and Group 4). We can obtain more specific findings with studies with large case series that group only the patients who have undergone the LEEP procedure.

## 5. Conclusions

In conclusion, the study findings indicated that HPV 16 and 18 that had positivity along with abnormal cytological findings are critical clinical situations that cause higher anxiety levels in patients infected with HPV. Despite the fact that sexual dysfunction was observed in HPV-infected women to a varying extent, the association between sexual dysfunction and the type of HPV positivity and cytological findings could not be elucidated. Thus, further studies with larger samples and more extended follow-up periods are needed. Raising public awareness about HPV by health professionals, including the HPV vaccine in the national vaccination program, and facilitating access to the treatment of cervical pathologies, will lead to less psychological strain on women, and therefore, less damage to the sexual lives of couples.

## Figures and Tables

**Figure 1 ijerph-20-03630-f001:**
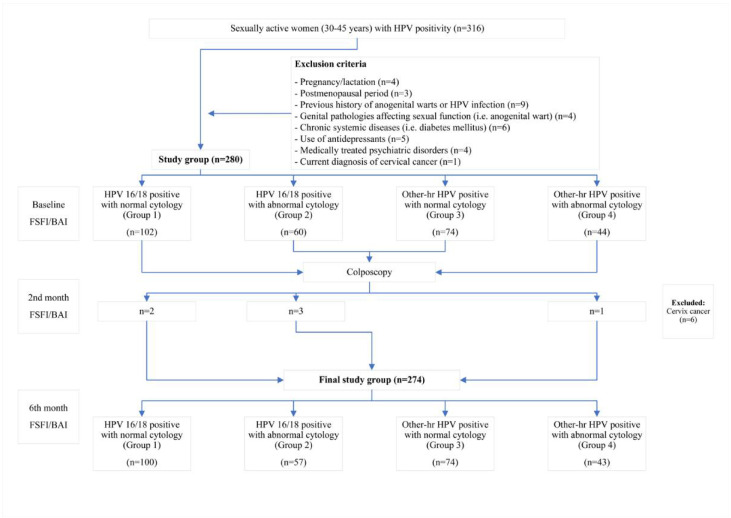
Flowchart of the study.

**Figure 2 ijerph-20-03630-f002:**
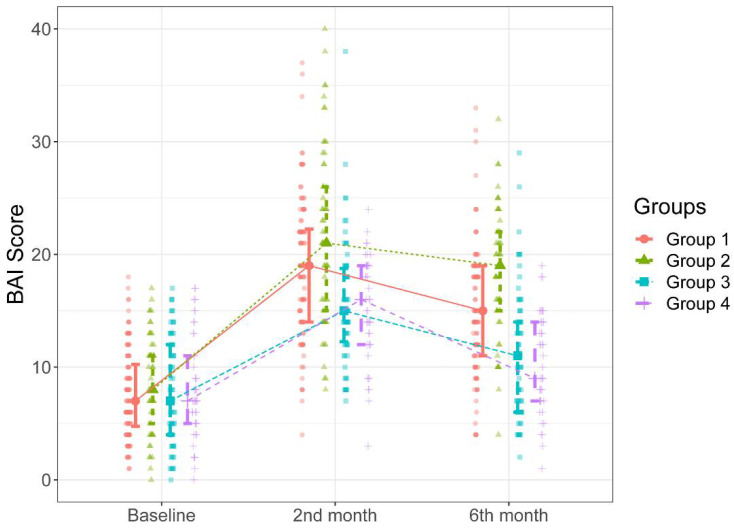
BAI scores of the groups during the study intervals.

**Figure 3 ijerph-20-03630-f003:**
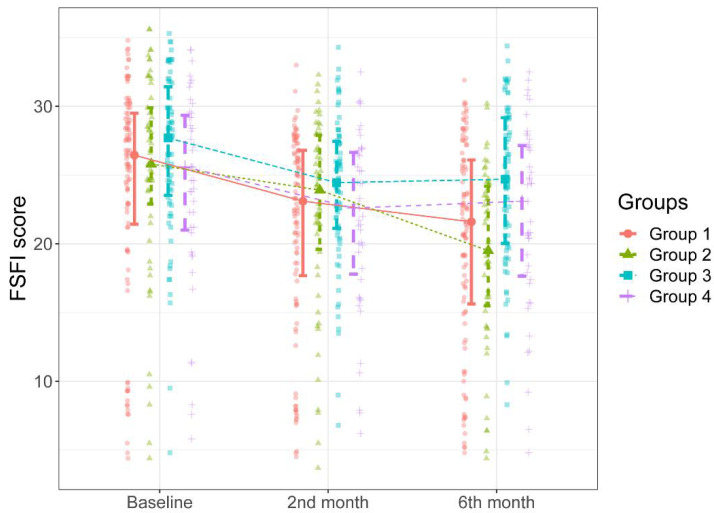
FSFI scores of the groups during the study intervals.

**Table 1 ijerph-20-03630-t001:** Characteristics of the study groups based on the HPV status and cytological findings.

	Group 1 (n = 100)	Group 2 (n = 57)	Group 3 (n = 74)	Group 4 (n = 43)
HPV				
HPV 16/18	80 (80.0)	38 (66.7)	0 (0.0)	0 (0.0)
Other-hr HPV	20 (20.0)	19 (33.3)	74 (100.0)	43 (100.0)
HPV genotypes				
HPV 16	51 (51.0)	25 (43.9)	0 (0.0)	0 (0.0)
HPV 18	29 (29.0)	13 (22.8)	0 (0.0)	0 (0.0)
Other-hr HPV	0 (0.0)	0 (0.0)	74 (100.0)	43 (100.0)
Multiple	20 (20.0)	19 (33.3)	0 (0.0)	0 (0.0)
Cytology				
Normal	100 (100.0)	0 (0.0)	74 (100.0)	0 (0.0)
LGSIL	0 (0.0)	15 (26.3)	0 (0.0)	14 (32.6)
HGSIL	0 (0.0)	25 (43.9)	0 (0.0)	7 (16.3)
ASC-US	0 (0.0)	17 (29.8)	0 (0.0)	22 (51.2)
Pathology				
Normal	34 (34.0)	4 (7.0)	41 (55.4)	17 (39.5)
CIN1	34 (34.0)	14 (24.6)	20 (27.0)	17 (39.5)
CIN2	18 (18.0)	23 (40.4)	12 (16.2)	6 (14.0)
CIN3	14 (14.0)	16 (28.1)	1 (1.4)	3 (7.0)
LEEP	30 (30.0)	39 (68.4)	17 (23.0)	12 (27.9)

HPV: human papilloma virus; LGSIL: low-grade intraepithelial squamous lesion; HGSIL: high-grade intraepithelial squamous lesion; ASC-US: atypical squamous cells of undetermined significance; CIN: cervical intraepithelial neoplasia; LEEP: loop electrosurgical excision.

**Table 2 ijerph-20-03630-t002:** Demographic and clinical characteristics of the study groups.

	Group 1 (n = 100)	Group 2 (n = 57)	Group 3 (n = 74)	Group 4 (n = 43)	*p*
Age (year)	39.0 [30.0–45.0]	38.0 [30.0–45.0]	39.0 [30.0–45.0]	39.0 [31.0–45.0]	0.506 **
Gravida	2.0 [0.0–6.0]	2.0 [0.0–4.0]	2.0 [0.0–5.0]	1.0 [0.0–4.0]	0.347 **
Smoking	25 (25.0)	27 (47.4)	21 (28.4)	15 (34.9)	0.029 *
Contraception					
No contraception	52 (52.0)	34 (59.6)	43 (58.1)	24 (55.8)	0.780 *
Contraception method	48 (48.0)	23 (40.4)	31 (41.9)	19 (44.2)	
IUD	19 (39.6)	7 (30.4)	14 (45.2)	3 (15.8)	0.229 *
Pills	7 (14.6)	3 (13.0)	6 (19.4)	4 (21.1)	
Bilateral tube ligation	8 (16.7)	1 (4.3)	2 (6.5)	2 (10.5)	
Caps/diaphragms	14 (29.2)	10 (43.5)	9 (29.0)	9 (47.4)	
Others	0 (0.0)	2 (8.7)	0 (0.0)	1 (5.3)	

IUD: intrauterine device, *. Pearson chi-Square or Fisher–Freeman–Halton test; **. Kruskal–Wallis H test.

**Table 3 ijerph-20-03630-t003:** Comparison between the study groups with respect to anxiety and sexual function scores and their changes over time.

	Group 1 (n = 100)	Group 2 (n = 57)	Group 3 (n = 74)	Group 4 (n = 43)	*p* *
BAI					
Baseline	7.0 [1.0–18.0]	8.0 [0.0–17.0]	7.0 [0.0–17.0]	7.0 [0.0–17.0]	0.514
2nd month	19.0 [4.0–37.0]	21.0 [8.0–40.0]	15.0 [7.0–38.0]	16.0 [3.0–24.0]	<0.001
6th month	15.0 [4.0–33.0]	19.0 [4.0–32.0]	11.0 [2.0–29.0]	9.0 [1.0–19.0]	<0.001
*p*’	<0.001	<0.001	<0.001	<0.001	
∆ 2nd month baseline (%)	145.0 [0.0–2700.0]	160.7 [0.0–1200.0]	100.0 [0.0–2000.0]	100.0 [0.0–1200.0]	0.029
∆ 6th month baseline (%)	88.8 [−25.0–1900.0]	104.5 [0.0–1000.0]	25.0 [−29.4–900.0]	18.5 [−50.0–600.0]	<0.001
FSFI total					
Baseline	26.4 [4.4–34.8]	25.8 [4.4–35.6]	27.7 [4.8–35.3]	25.6 [5.8–34.1]	0.275
2nd month	23.1 [4.5–33.0]	23.9 [3.7–32.3]	24.4 [6.8–34.3]	22.6 [6.2–32.5]	0.181
6th month	21.6 [4.8–31.9]	19.5 [4.4–30.2]	24.7 [8.3–34.4]	23.1 [4.8–32.5]	<0.001
*p*’	<0.001	<0.001	<0.001	<0.001	
∆ 2nd month baseline (%)	−11.5 [−30.4–11.4]	−11.0 [−32.8–0.3]	−9.1 [−27.8–41.7]	−9.6 [−29.2–6.9]	0.066
∆ 6th month baseline (%)	−11.8 [−50.4–18.2]	−27.3 [−49.8–16.4]	−2.6 [−41.5–72.9]	−2.4 [−42.2–21.0]	<0.001
Desire					
Baseline	4.2 [1.2–6.0]	4.2 [1.2–6.0]	4.2 [1.2–6.0]	3.6 [1.2–5.4]	0.160
2nd month	3.0 [1.2–5.4]	3.6 [1.2–4.8]	3.0 [1.2–5.4]	2.4 [1.2–5.4]	0.091
6th month	3.0 [1.2–4.8]	3.0 [1.2–4.8]	3.6 [1.2–5.4]	3.6 [1.2–5.4]	<0.001
∆ 2nd month baseline (%)	−22.2 [−71.4–0.0]	−16.7 [−60.0–0.0]	−25.0 [−71.4–0.0]	−25.0 [−75.0–16.7]	0.278
∆ 6th month baseline (%)	−16.7 [−75.0–50.0]	−28.6 [−75.0–50.0]	0.0 [−66.7–50.0]	0.0 [−62.5–50.0]	<0.001
Arousal					
Baseline	4.5 [0.0–6.0]	4.2 [0.0–6.0]	4.5 [0.0–5.7]	4.2 [0.0–5.4]	0.311
2nd month	3.3 [0.0–5.4]	3.6 [0.0–4.5]	3.3 [0.0–5.7]	3.0 [0.0–4.8]	0.355
6th month	3.3 [0.0–5.1]	3.0 [0.6–5.1]	4.2 [0.6–5.7]	3.9 [0.0–5.7]	<0.001
∆ 2nd month baseline (%)	−21.8 [−66.7–0.0]	−17.2 [−61.1–0.0]	−20.0 [−55.6–0.0]	−24.3 [−75.0–16.7]	0.806
∆ 6th month baseline (%)	−15.4 [−72.7–40.0]	−28.6 [−76.9–0.0]	0.0 [−62.5–20.0]	0.0 [−58.8–50.0]	<0.001
Lubrication					
Baseline	4.5 [0.0–6.0]	4.2 [0.0–6.0]	4.5 [1.2–6.0]	4.2 [1.2–5.7]	0.204
2nd month	4.2 [0.0–6.0]	3.9 [0.0–5.7]	4.5 [1.2–6.0]	4.2 [1.2–5.7]	0.030
6th month	3.3 [0.0–6.0]	3.0 [0.0–5.4]	4.2 [1.5–5.7]	3.9 [0.0–6.0]	<0.001
∆ 2nd month baseline (%)	−6.2 [−26.7–0.0]	−7.1 [−25.0–0.0]	0.0 [−12.5–7.1]	0.0 [−15.4–7.1]	<0.001
∆ 6th month baseline (%)	−14.3 [−100.0–0.0]	−33.3 [−100.0–0.0]	0.0 [−70.0–25.0]	0.0 [−100.0–25.0]	<0.001
Orgasm					
Baseline	4.8 [0.0–6.0]	4.8 [0.4–6.0]	4.8 [1.2–6.0]	4.8 [0.4–6.0]	0.274
2nd month	4.4 [0.0–6.0]	4.4 [0.4–6.0]	4.8 [1.2–6.0]	4.4 [0.4–6.0]	0.025
6th month	4.0 [0.0–6.0]	3.6 [0.0–6.0]	4.4 [1.2–6.0]	4.0 [0.0–6.0]	0.001
∆ 2nd month baseline (%)	−7.1 [−60.0–10.0]	−8.3 [−57.1–7.7]	0.0 [−33.3–7.1]	0.0 [−25.0–10.0]	<0.001
∆ 6th month baseline (%)	−9.1 [−71.4–0.0]	−21.4 [−100.0–0.0]	0.0 [−72.7–11.1]	0.0 [−100.0–20.0]	<0.001
Satisfaction					
Baseline	4.8 [0.8–6.0]	4.8 [0.8–6.0]	5.2 [1.2–6.0]	4.8 [0.8–6.0]	0.234
2nd month	4.8 [0.8–6.0]	4.8 [0.8–6.0]	4.8 [1.6–6.0]	4.8 [0.8–5.6]	0.071
6th month	4.4 [0.8–6.0]	4.4 [0.8–6.0]	4.8 [1.6–6.0]	4.8 [0.8–6.0]	0.001
∆ 2nd month baseline (%)	−7.1 [−33.3–8.3]	0.0 [−33.3–8.3]	0.0 [−33.3–33.3]	0.0 [−25.0–33.3]	0.079
∆ 6th month baseline (%)	−7.7 [−57.1–10.0]	−15.4 [−75.0–8.3]	0.0 [−55.6–33.3]	0.0 [−62.5–33.3]	<0.001
Pain					
Baseline	4.0 [0.0–5.6]	4.0 [0.0–6.0]	4.0 [0.0–6.0]	4.0 [1.2–5.6]	0.483
2nd month	3.6 [0.0–5.6]	4.0 [0.0–6.0]	4.0 [1.6–6.0]	3.6 [1.2–6.0]	0.411
6th month	3.6 [0.0–5.6]	3.6 [0.0–5.2]	3.6 [1.2–6.0]	3.6 [1.2–5.6]	0.504
∆ 2nd month baseline (%)	0.0 [−25.0–33.3]	0.0 [−14.3–7.7]	0.0 [−20.0–33.3]	0.0 [−50.0–33.3]	0.836
∆ 6th month baseline (%)	0.0 [−62.5–33.3]	−7.4 [−62.5–0.0]	0.0 [−60.0–33.3]	0.0 [−76.9–66.7]	0.023

BAI: Beck Anxiety Inventory, FSFI: Female Sexual Function Index. *: Kruskal Wallis H test, ‘: Friedman test.

**Table 4 ijerph-20-03630-t004:** Comparison between the groups based on the presence of HPV 16/18 positivity with respect to anxiety and sexual function scores and their changes over time.

	Groups with HPV 16/18 (Group 1 and Group 2) (n = 157)	Groups with Other-hr HPV (Group 3 and Group 4) (n = 117)	*p* *
BAI			
Baseline	8.0 [0.0–18.0]	7.0 [0.0–17.0]	0.801
2nd month	19.0 [4.0–40.0]	15.0 [3.0–38.0]	<0.001
6th month	16.0 [4.0–33.0]	10.0 [1.0–29.0]	<0.001
∆ 2nd month baseline	150.0 [0.0–2700.0]	100.0 [0.0–2000.0]	0.004
∆ 6th month baseline	100.0 [−25.0–1900.0]	20.0 [−50.0–900.0]	<0.001
FSFI total			
Baseline	26.4 [4.4–35.6]	26.5 [4.8–35.3]	0.432
2nd month	23.3 [3.7–33.0]	24.1 [6.2–34.3]	0.256
6th month	21.2 [4.4–31.9]	24.4 [4.8–34.4]	<0.001
∆ 2nd month baseline	−11.2 [−32.8–11.4]	−9.2 [−29.2–41.7]	0.012
∆ 6th month baseline	−15.2 [−50.4–18.2]	−2.5 [−42.2–72.9]	<0.001
Desire			
Baseline	4.2 [1.2–6.0]	4.2 [1.2–6.0]	0.568
2nd month	3.0 [1.2–5.4]	3.0 [1.2–5.4]	0.738
6th month	3.0 [1.2–4.8]	3.6 [1.2–5.4]	<0.001
∆ 2nd month baseline	−20.0 [−71.4–0.0]	−25.0 [−75.0–16.7]	0.183
∆ 6th month baseline	−20.0 [−75.0–50.0]	0.0 [−66.7–50.0]	<0.001
Arousal			
Baseline	4.2 [0.0–6.0]	4.2 [0.0–5.7]	0.412
2nd month	3.3 [0.0–5.4]	3.3 [0.0–5.7]	0.719
6th month	3.3 [0.0–5.1]	3.9 [0.0–5.7]	<0.001
∆ 2nd month baseline	−20.0 [−66.7–0.0]	−21.4 [−75.0–16.7]	0.908
∆ 6th month baseline	−21.1 [−76.9–40.0]	0.0 [−62.5–50.0]	<0.001
Lubrication			
Baseline	4.5 [0.0–6.0]	4.5 [1.2–6.0]	0.373
2nd month	4.2 [0.0–6.0]	4.2 [1.2–6.0]	0.020
6th month	3.3 [0.0–6.0]	4.1 [0.0–6.0]	<0.001
∆ 2nd month baseline	−6.7 [−26.7–0.0]	0.0 [−15.4–7.1]	<0.001
∆ 6th month baseline	−20.0 [−100.0–0.0]	0.0 [−100.0–25.0]	<0.001
Orgasm			
Baseline	4.8 [0.0–6.0]	4.8 [0.4–6.0]	0.477
2nd month	4.4 [0.0–6.0]	4.8 [0.4–6.0]	0.010
6th month	4.0 [0.0–6.0]	4.4 [0.0–6.0]	0.001
∆ 2nd month baseline	−7.7 [−60.0–10.0]	0.0 [−33.3–10.0]	<0.001
∆ 6th month baseline	−15.4 [−100.0–0.0]	0.0 [−100.0–20.0]	<0.001
Satisfaction			
Baseline	4.8 [0.8–6.0]	5.2 [0.8–6.0]	0.311
2nd month	4.8 [0.8–6.0]	4.8 [0.8–6.0]	0.125
6th month	4.4 [0.8–6.0]	4.8 [0.8–6.0]	0.001
∆ 2nd month baseline	−7.1 [−33.3–8.3]	0.0 [−33.3–33.3]	0.024
∆ 6th month baseline	−9.1 [−75.0–10.0]	0.0 [−62.5–33.3]	<0.001
Pain			
Baseline	4.0 [0.0–6.0]	4.0 [0.0–6.0]	0.278
2nd month	3.6 [0.0–6.0]	4.0 [1.2–6.0]	0.323
6th month	3.6 [0.0–5.6]	3.6 [1.2–6.0]	0.170
∆ 2nd month baseline	0.0 [−25.0–33.3]	0.0 [−50.0–33.3]	0.419
∆ 6th month baseline	0.0 [−62.5–33.3]	0.0 [−76.9–66.7]	0.037

*: Mann-Whitney U test.

**Table 5 ijerph-20-03630-t005:** Correlation analysis of BAI and total FSFI scores between the groups based on the HPV 16/18 positivity and abnormal cytological findings.

	Group 1		Group 2		Group 3		Group 4	
	r	*p*	r	*p*	r	*p*	r	*p*
2nd month	−0.116	0.251	0.101	0.453	0.045	0.705	−0.396	0.009
6th month	−0.213	0.033	−0.075	0.581	−0.375	<0.001	−0.605	<0.001
	**Group 1 and Group 2 (HPV 16/18)**				**Group 3 and Group 4 (Other hr-HPV)**			
	**r**		** *p* **		**r**		** *p* **	
2nd month	−0.029		0.718		−0.131		0.160	
6th month	−0.198		0.013		−0.439		<0.001	
	**Group 1 and Group 3 (Normal Cytology)**				**Group 2 and Group 4 (Abnormal Cytology)**			
	**r**		** *p* **		**r**		** *p* **	
2nd month	−0.109		0.153		−0.070		0.489	
6th month	−0.338		<0.001		−0.353		<0.001	

## Data Availability

Data sharing is not applicable to this article.

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
