# Peer review of "The Impact of HPV Diagnosis and Abnormal Cervical Cytology Results on Sexual Dysfunction and Anxiety"

_ijerph, 2023, doi:10.3390/ijerph20043630_

Round 1
Reviewer 1 Report
The researchers investigated an interesting issue.
I think that it is adequate for publication.
I have a suggestion for the paper.
''Comba C, Demirayak G, Erdogan SV, Karaca I, Demir O, Guler O, Ozdemir IA. Comparison of pain and proper sample status according to usage of tenaculum and analgesia: a randomized clinical trial. Obstet Gynecol Sci. 2020 Jul;63(4):506-513. doi: 10.5468/ogs.19185. Epub 2020 Jun 19. PMID: 32550738; PMCID: PMC7393752.'' this article should cite for how colposcopy performed. Because negative impact of the diagnostic procedure can be on sexual behavior.
Author Response
We added information about preventing the effect of the LEEP procedure on patients' pain perception and sexual function to discussion section and cited the publication you suggested.
“The LEEP procedure was performed under local anesthesia as stated in Comba et al.'s study to eliminate the negative impact on patients' pain experience and sexual functions [26]”
Reviewer 2 Report
Dear authors: It is required to include in the methodology section the ethical processes that should be applied in this research, such as: approval of the ethics committee and signing of informed consent.
Best regards.
Author Response
According to the rules of the journal, information about the ethics committee and patient consent should be given at the end of the discussion section, not in the material and method section. We regret that we were unable to fulfill your request.
Ref: https://www.mdpi.com/journal/ijerph/instructions#preparation
Reviewer 3 Report
The Impact of HPV Diagnosis and Abnormal Cervical Cytology Results on Sexual Dysfunction and Anxiety
This is a very interesting and important contribution to the field of female sexuality and female sexual health, with important implications for clinical practice and prevention. Still, I believe that a few changes in the manuscript would increase its chances of being published:
1. The introduction is scarce and lacks further descriptions of more previous research done in this field. Please include more references and provide further information on the topics of female sexual dysfunction and anxiety symptoms, as well as risk and protective factors.
2. Objectives must be clearly and thoroughly explained.
3. Please provide more explanation on 2.1.
4. Cultural aspects of Turkey and female sexuality must be operationalized to contextualize access to HPV testing, respective barriers and difficulties.
5. Please provide more information about the BAI and the FSFI measurement instruments, namely, reliability information (Cronbach’s alpha).
6. Figures 2 and 3 must be more clearly explained in the text.
7. The study had more limitations than the ones presented in line331; please provide a list of more detailed and explained limitations. Please write them in a sub section of the discussion section.
8. Please provide an implications section of the study’s results, focusing on implications for clinical practice, prevention, health promotion, social and political policies, etc.
Best wishes.
Author Response
- The introduction is scarce and lacks further descriptions of more previous research done in this field. Please include more references and provide further information on the topics of female sexual dysfunction and anxiety symptoms, as well as risk and protective factors.
- The introduction part is detailed with new references. And these sentences has been added to the introduction section.
“Female patients who receive positive HPV test results are reported to experience anxiety and worry [8]. Similar to this, Lin et al's study revealed that HPV patients encounter a number of unfavorable feelings, primarily dread and anxiety [12]. Another important point to be emphasized about HPV transmission is the negative effects on female sexuali-ty. Ferenidau et al. reported that after being diagnosed with HPV infection, women had less sexual interest and had less sexual intercourse [13]. Similarly, in the study of Lin et al., after interviewing women infected with high-risk HPV genotypes, these women reported that they had infrequent sexual intercourse after HPV diagnosis [12]. Likewise, McCaffery et al. reported that women diagnosed with HPV hesitate to share their test results with their partners and have negative feelings about their sexual life. This may be due to fear of infecting the partner, fear of rejection, guilt, and anger at having been betrayed [8].”
- Objectives must be clearly and thoroughly explained.
- The objectives were stated in more detail. These sentences below has been added to the introduction section.
“The aim of this study to investigate the effects of different HPV types and cervical cytology results on sexual functions and anxiety levels of Turkish women. Furthermore, we wanted to determine which subgroup of sexual functions is mostly affected by different HPV genotypes and cervical cytology results.”
- Please provide more explanation on 2.1.
- More clarification has been made about section 2.1. The sentence has been changed.
“This study was designed as a prospective observational study to evaluate the effect of HPV diagnosis, HPV types and cervical cytology abnormalities on sexual function and anxiety levels of Turkish women.”
- More detailed information about the study is available in section 2.2. If you still think that section 2.1 is insufficient, we can transfer some information from 2.2 to 2.1.
- Cultural aspects of Turkey and female sexuality must be operationalized to contextualize access to HPV testing, respective barriers and difficulties.
- In Turkey, HPV awareness among women has been increasing in recent years. The national cervical cancer screening program is widely applied to all women over the age of 30. The majority of women are reached through community health centers, and HPV testing and cervical cytology are applied free of charge in all public hospitals. There are no restrictions on women's access to HPV testing. The sentence has been added below the 2.2 section.
“The majority of women are reached through community health centers, and HPV testing and cervical cytology are applied free of charge in all public hospitals.”
- Please provide more information about the BAI and the FSFI measurement instruments, namely, reliability information (Cronbach’s alpha).
- Data on the reliability and validity of these tests were added to the title of BAI scores. “The BAI showes a high internal consistency (Cronbach’s alpha= 0.93). The item-total correlations ranged from 0.45 to 0.72.”
- Data on the reliability and validity of these tests were added to the title of FSFI scores.
“The internal consistency of the FSFI test is high, (Cronbach’s alpha = 0.83-0.96) and the test-retest reliability is ranged from 0.74 to 0.86.”
- Figures 2 and 3 must be more clearly explained in the text.
- Explanatory sentences about Figure 2 are given. Sentences to point to Figure 2 have been added to the paragraph.
“In Figure 2, it is seen that there were significant increases in the BAI scores over time compared to the baselines scores in all groups (p=0.001 for all). The patients in Groups 1 and 2 had significantly higher BAI scores measured during the second-month follow-up than those in Groups 3 and 4 (p<0.05). There were no significant differences between Group 1 and Group 2 and between Group 3 and 4 in BAI scores measured during the two-month follow-up (p>0.05).”
“Likewise, as can be seen in Figure 2 The patients in Group 2 had significantly higher BAI scores measured during the six-month follow-up than the patients in other groups (p<0.001 for all cases). Additional-ly, the patients in Group 1had significantly higher BAI scores measured during the six-month follow-up than those in Group 3 and Group 4 (p<0.001 for both cases). There was no significant difference between Group 3 and Group 4 in BAI scores measured dur-ing the six-month follow-up(p=0.983). The median increases in the BAI scores between the two-month and six-month follow-ups were higher in Groups 2 and 1 than those in Groups 3 and 4 (Table 3, Figure 2).”
- Explanatory sentences about Figure 3 are given. Sentences to point to Figure 3 have been added to the paragraph.
“As can be seen in figure 3, There were significant progressive decreases in patients’ total FSFI scores over time compared to the baseline scores in Groups1 and 2 (p<0.001 and p<0001). There were no significant differences between the groups in total FSFI scores determined during the sec-ond-month follow-up (p=0.181).”
“Figure 3 shows that the total FSFI scores determined in the six-month follow-up of the patients in Group 1 and 2 were significantly lower than those in Group 3 (p=0.004 and p<0.001, respectively). The total FSFI score determined during the six-month follow-up was lower in Group 2 than in Group 1, albeit not statistically significant (p=0521). The median percent decrease in the total FSFI score determined during the six-month follow-up compared to the total FSFI scores determined before was higher in Group 2 than in the other groups (Table 3, Figure 3).”
- The study had more limitations than the ones presented in line331; please provide a list of more detailed and explained limitations. Please write them in a sub section of the discussion section.
- More limitations has been added to the end of the discussion section.
“If there were more cases included in our study, we could obtain statistically more dis-criminating results. We could follow the patients for more than 6 months. But under the conditions of our country, we have difficulty in reaching all of these cases whenever we want. If we had extended it to 1 year, we would have had problems with the continuity of the patients. It is controversial whether the LEEP procedure has an effect on sexual func-tions. The fact that the LEEP procedure performed in the HPV 16-18 group was higher than the other group may be a negative impact factor for sexual dysfunctions (43% of Group 1 and Group 2 ; 24% of Group 3 and Group 4). We can obtain more specific find-ings with studies with large case series that group only the patients who have undergone the LEEP procedure.”
- Please provide an implications section of the study’s results, focusing on implications for clinical practice, prevention, health promotion, social and political policies, etc.
- This sentence has been added to the end of the conclusion section.
“Raising public awareness about HPV by health professionals, including the HPV vaccine in the national vaccination program, and facilitating access to the treatment of cervical pathologies will lead to less psychological destruction of women and therefore less damage to the sexual lives of couples.”
Round 2
Reviewer 3 Report
Thank you very much for implementing all the requested changes.
I believe the manuscript has been sufficiently improved to warrant publication in IJERPH.
Best wishes.